

# Divergence of compost extract and bio-organic manure effects on lucerne plant and soil

Haiyan Ren[1], Jian Hu[1], Yifei Hu[1], Gaowen Yang[1] and Yingjun Zhang[1,2]

[1] College of Agro-grassland Science, Nanjing Agricultural University, Nanjing, China
[2] Department of Grassland Science, China Agricultural University, Beijing, China

## ABSTRACT

**Aim**. Application of organic materials into agricultural systems enhances plant growth and yields, and improves soil fertility and structure. This study aimed to examine the effects of "compost extract (CE)", a soil conditioner, and bio-organic manure (BOM) on the growth of lucerne (*Medicago sativa*), and compare the efficiency between BOM (including numbers of microorganisms) and CE (including no added microorganisms).
**Method**. A greenhouse experiment was conducted with four soil amendment treatments (control, BOM, CE and CEBOM), and was arranged in a completely randomized design with 10 replicates for each treatment. Plant biomass, nutritive value and rhizobia efficacy as well as soil characteristics were monitored.
**Result**. CE rather than BOM application showed a positive effect on plant growth and soil properties when compared with the control. Lucerne nodulation responded equally to CE addition and rhizobium inoculation. CE alone and in combination with BOM significantly increased plant growth and soil microbial activities and improved soil structure. The synergistic effects of CE and BOM indicate that applying CE and BOM together could increase their efficiency, leading to higher economic returns and improved soil health. However, CE alone is more effective for legume growth since nodulation was suppressed by nitrogen input from BOM. CE had a higher efficiency than BOM for enriching soil indigenous microorganisms instead of adding microorganisms and favouring plant nodulation.

## INTRODUCTION

Preserving and restoring native grasslands and agricultural production systems by improving soil quality have been common goals worldwide (*Henwood, 2010*). However, long-term inappropriate management and large inputs of chemical fertilizers and pesticides into soils have led to severe soil problems. Commercial soil conditioners or manures are used in agricultural production systems for amending nutrient-deficient soils by providing multiple nutrients (*Soumare, Tack & Verloo, 2003*; *Hu & Qi, 2013*), including, but not limited to, bio-organic manure (BOM), effective microorganisms (EM), and compost extract (CE). However, farmers are sometimes reluctant to use organic fertilizers because the effects of organic nutrients on plant growth are not as quickly seen as with inorganic

Corresponding author
Haiyan Ren, hren@njau.edu.cn

fertilizers. Thus, new materials and new techniques for increasing the efficiency of organic materials are needed.

Introducing exogenous microorganisms into agricultural environments has been applied to accelerate bioremediation and improve agricultural productivity (*Paluch, Thomsen & Volk, 2013*). Although less is known about the practical effectiveness and mechanisms of these products, the assessment of their impacts has focused on their potential to increase soil organic matter content, promote nutrient availability to crops (*Khan, Zaidi & Wani, 2007*), enhance the proliferation of beneficial bacteria (*Priyadi et al., 2005*; *Javaid, 2010*), and prevent infection by pathogens (*Compant et al., 2005*). However, additional microorganisms from commercial products have to be incubated formerly and propagated in the lab and further applied to the fields. Their survival rate and their influences on plant and soil have been viewed skeptically (*Watanabe, Miyashita & Harayama, 2000*; *Chen et al., 2015*).

In contrast, indigenous microorganisms may be optimally suited for survival and multiplication in significant amounts (*Paluch, Thomsen & Volk, 2013*). However, there is less information available on the ways to activate naturally occurring soil microbial populations. The use of CE produced by plant extraction technology may be one method to activate indigenous soil microbial populations. CE derived from endophytes extracted from different plant species such as soybean (*Glycine max* (L.) Merr.), lucerne (*Medicago sativa* L.), the brown algae (*Ascophyllum nodosum*) (Bestfarming systems Co.), and has been widely used recently. A new bio-molecular method for determining the presence of *nifH*, the gene harboured in endophytic bacteria for nitrogenase reductase from nodulated plant species has been developed (http://www.amazingcarbon.com) (*Gao et al., 2015*; *Le et al., 2015*). This study demonstrates that CE activates indigenous microorganisms in the soil, particularly N-fixing bacteria or archaea that are not able to be cultured in the laboratory. Besides, increasing evidence suggests that is of lavonoids and flavonoids exuded from the root of many legumes can activate rhizobium genes, which helps in the nodulation process (*Peters, Frost & Long, 1986*; *Brunetti et al., 2013*). Combing microbial remediation with legume remediation is expected to be an economically and environmentally appropriate approach for soil amendment.

This study was undertaken to verify if: (1) there are positive effects of adding CE on plant and microbial biomass, microbial activity, plant and soil nutrient contents; (2) combining CE and BOM achieve the highest impacts on soil nutrient contents and lucerne growth; (3) CE stimulates the nodulation of legume as effectively as by inoculating rhizobacteria. Results are expected to reveal the interactive effects of CE and BOM application on plant species growth and soil properties, and aid in the selection of optimal fertilization approaches for soil quality improvement. In addition, the experiments will enable exploring the stimulative effect of CE on activating and enriching indigenous microorganisms from plant rhizosphere.

## MATERIALS AND METHODS

### Experimental design

A greenhouse experiment was conducted in the laboratory of Prataculture Science, Nanjing Agriculture University, from June 2015 to September 2015 as preliminary trial with only

**Table 1  Soil properties in the study area.**

| Soil property | pH | Bulk density (g/cm³) | WHC (%) | OM (g/kg) | Organic C (%) | Total N (%) | Total P (%) | C: N ratio (%) |
|---|---|---|---|---|---|---|---|---|
| Value | 5.88 | 1.61 | 25.73 | 13.25 | 12.12 | 0.83 | 0.16 | 14.60 |

**Notes.**

OM, organic matter; WHC, water holding capacity.

visual evaluation of treatment effects. To determine the actual effects of the treatments, the experiment was completely repeated from October 2015 to January 2016, with quantitative data recorded. The experiment was set up with four soil amendment treatments and 10 replications each. The four treatments included: bio-organic manure (BOM), compost extract (CE), organic manure + compost extract (CEBOM) and the control.

We collected lucerne seed from barren land in Xilinhot (43°38′N, 116°42′E), Inner Mongolia, China. The seed were sterilized firstly and germinated on 1.5% water-agar plates and transplanted into pots (height 15 cm, diameter 15 cm). Each pot was filled with 1 kg of soil (dry weight equivalent) and 200 g double-washed quartz sand. Soil without any heavy metal content was collected from the same study area and the soil had 1-year history for planting legumes. The soil was collected to 15 cm depth with 5 cm diameter soil cores and evenly mixed, then applied to each pot. The characteristics of the soil are listed in Table 1. Prior to seedling transplantation, BOM was collected and mixed into the top layer of the potting at a rate of 15 g per pot. It was made of 50% pig dung compost and 50% amino acid fertilizer; the pig dung compost made by composting pig dung at 30 to 70 °C for 25 days, contained 30.1% organic matter, 3.0% N, 2.7% $P_2O_5$, 0.9% $K_2O$; the amino acid fertilizer enzymatically hydrolyzed by aerobic microbial fermeantation for 7 days, contained 40.2% organic matter, 11.1% amino acids, 3.4% N, 1.7% $P_2O_5$, 1.1% $K_2O$ (*Zhang et al., 2011*; *Huang et al., 2011*). Four of ten strongest seedlings were finally left per pot. CE consisted of a mixture alfalfa meal 15.9% (weight/weight), barley grain 10.2%, barley straw 6.4%, wheat straw 4.3%, liquidized fish 8.7%, kelp 39.5%, sulphur 0.3%, calcium carbonate 10.2%, and molasse 4.5%. The mixture was fermented at <50 °C; for 14 days in an incubator, then extracted, filtered, freezed and dried. The final compositions (minimum guaranteed analysis) contained 16.9% amino acids, 0.5% soluble potash (weight/weight), 0.06% calcium, 1.5% sulphur, 1.2% nitrogen, 0.3% phosphorus. 20 ml of CE (Bestfarming Systems Co.) (2 ml/pot for each application) diluted with 5 L of water were sprayed onto the soil at a rate comparable to agricultural practice with 750 ml /hectare. Four weeks later, CE was applied for a second time (practical dosage at manufacturers' recommendation) and no more after that.

A supplementary experiment on nodulation effects was conducted when seedlings were transplanted into additional pots (same size as above). The seedlings in the pots were inoculated with 6 ml liquid per pot with rhizobium strains (*Rhizobium melilotid* Dangeard-CX 107) cultured from nodules (provided by Key Laboratory of Agro-Microbial Resource and Application, China Agriculture University).Bacterial suspensions were diluted to an optical density of 0.70 ($\lambda = 600$ nm), which was equivalent to a concentration of $3.10^8$
bacteria/ml, as measured by Bradford method (*Bradford, 1976*) on a yeast-mannitol culture medium (*Vincent & Humphrey, 1970*).

Starting from 10. June 2015, seedlings were grown under glasshouse conditions at 50–70% relative humidity and a temperature regime of 20–25 °C during day and night. Pots were watered to maintain soil water content between 20 and 25%. Eighty days later, seedlings of both experiments were destructively harvested, and separated into leaves, shoots and roots, their biomass measured, and were examined for nodulation by counting and weighing root nodules (roots were washed to remove adhering soil before checking).

## Chemical analyses
### Plant samples

Samples of stem and leaves were milled to <0.2 mm and analyzed for total C and N. Plant total P content was determined using a UV/visible spectrophotometer (Beckman Coulter DU 800, USA). Plant total K content was measured using $H_2SO_4$-salicylic acid -$H_2O_2$-Se digestion (*Zheng et al., 2012*).

Plant samples were ground to pass a 1 mm sieve and scanned twice using Near-Infrared-Spectroscopy (NIRS) technique for forage nutritive value determination (*Ren et al., 2016*). Nutritional value was tested by evaluating crude protein (CP; nitrogen concentration × 6.25), neutral detergent fibre (NDF), cellulose digestible organic matter (CDOM) and metabolisable energy (ME) (MJ/kg DM) (*Schonbach et al., 2009*). NDF was analyzed sequentially using the method described by *Vansoest, Robertson & Lewis (1991)*, which used semiautomatic ANKOM 220 technology and was expressed with residual ash. CDOM value as a percentage of organic matter and ME value were calculated using crude ash (CA) and non-soluble enzymatic substance (EULOS). Detailed calculations are given in Eqs. (1) and (2) below (*Schonbach et al., 2009*):

$$CDOM = 100\,(940 - CA - 0.62\,EULOS - 0.000221\,EULOS^2)/(1000 - CA) \qquad (1)$$

$$ME = 13.98 - 0.0147\,CA - 0.0102\,EULOS - 0.00000254\,EULOS^2 + 0.00234\,CP \qquad (2)$$

### Soil samples

Soil samples were separated in two portions, one portion was weighed to determine soil bulk density by using a foil sampler (volume $=100\ cm^3$) and dried at 105 °C for 24 h to calculate water holding capacity (%), while another portion was stored at −22 °C for microbial abundance analysis by real-time PCR (*Gupta et al., 2017*). Subsamples were sieved to 0.15 mm and analyzed for soil organic matter (OM), alkaline hydrolysable nitrogen (N), available phosphorus (P), available potassium (K), and total C, N, P and K content. Soil OM content was determined through the potassium dichromate external heating method (*Sparks et al., 1996*). The alkaline-hydrolysable diffusion method was applied to determine alkaline- hydrolysable N content (*Bremner, Smith & Tarrant, 1996*). Available P was measured using a spectrophotometer, with ammonium molybdate and ascorbic acid as colour reagents (*Webster, 2008*). Available K was measured by ammonium nitrate extraction-the flame photometry method (*Sparks et al., 1996*). The organic C content was analyzed using $K_2Cr_2O_7$-$H_2SO_4$ oxidation method. Soil total N was analyzed with a Kjeltec

analyzer (Kjeltec Analyzer Unit 2300; FOSS, Hillerød, Sweden). Soil total P and K was measured in the same way as plant total P. Soil pH was measured in a 1:2.5 (soil: water) suspension.

Microbial biomass carbon (MBC) and microbial biomass nitrogen (MBN) as well as microbial respiration (MR) were measured to assess differences in microbial colonization and activity. MBC and MBN were estimated using the chloroform fumigation extraction method (*Frostegard, Tunlid & Baath, 2011*). MR was measured in pre-incubated (24 h at 28 °C) samples by determining $CO_2$ evolution over a period of 72 h (*Schlesinger & Andrews, 2000*). The abundance of viable bacteria and fungi as measured according to real-time quantitative PCR-DGGE (denaturing gradient gel electrophoresis) of 16S rDNA, which was performed using the ABI 7300 real-time PCR system (Applied Biosystems, Foster City, CA, USA) with fluorescence TaqMan@ probe detection (*Le et al., 2015*; *Gupta et al., 2017*).

## Statistical analysis

One-way ANOVA (analysis of variance) was used to determine differences among treatments for lucerne species growth and soil properties. The LSD-test procedure was used for testing mean differences among control, BOM and CE treatments. The level of significance was $P < 0.05$. All the variables were statistically tested for homogeneity of variance. All statistical analyses were carried out using SAS, Version 9.2 (SAS Institute Inc., Cary, NC, USA).

# RESULTS

## Biomass and nutritional value of lucerne

Shoot, root and total biomass of inoculated lucerne significantly increased ($P < 0.05$) under CE application by 73.3%, 38.3% and 63.8%, respectively (Fig. 1). Compared with the control, the three biomasses of lucerne did not change under CEBOM treatment, and decreased by17.5% to 26.6 % when BOM was added. For non-inoculated lucerne, all three biomasses showed highest values with CE addition alone by increasing from 46.0% to 85.3%, then were second with CEBOM treatment. BOM addition did not affect the biomass of non-inoculated lucerne. The effects of amendment treatments on soil characteristics and plant nutritional values were further analyzed by ANOVA (Table S1). CE and BOM significantly affected the nutritional values (Table 2). The CP concentration of lucerne was significantly higher in BOM and CE treatments than in the control. CDBOM and ME showed the same trend in response to treatments. CE and BOM significantly decreased lucerne NDF from 66.18 to 52.45 g kg$^{-1}$ DM and CE alone decreased NDF by 36.3%.

## N, P, K contents in stems, leaves and soil

The N content of stems, leaves and soil decreased significantly in the order CEBOM >BOM >CE >control (Table 3, Table S1). The N contents of stem, leaves and soil in the CEBOM treatment were 40.4%, 35.0% and 91.4% higher than in the control treatment, respectively. The P content of stems was significantly higher in the three soil amendment treatments than in the control treatment. The P content of leaves and soils increased in the order:

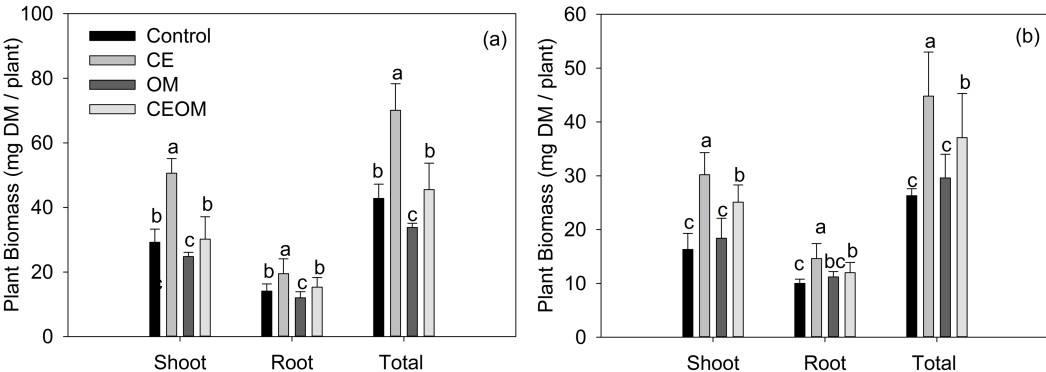

**Figure 1** Shoot biomass, root biomass and total biomass of inoculated (A) and non-inoculated (B) lucerne under different soil amendment treatments: control, compost extract (CE), bio-organic manure (BOM) and compost extract + bio-organic manure (CEBOM). Bars represent the standard errors, *n* = 20.

**Table 2** Effect of compost extract (CE), bio-organic manure (BOM) and CE + BOM (CEBOM) application on shoot nutrient values of lucerne.

| Treatment | CP (g DM/kg) | NDF (g DM/kg) | CDOM (g DM/kg) | ME (MJ DM/kg) |
|---|---|---|---|---|
| Control | 17.7c | 66.18a | 68.61b | 9.84b |
| CE | 23.3a | 58.24c | 77.77a | 10.87ab |
| BOM | 21.4b | 61.48ab | 72.42b | 10.09b |
| CEBOM | 24.6a | 52.45bc | 79.99a | 11.01a |

Notes.

DM, dry matter; CP, crude protein; NDF, neutral detergent fibre; CDOM, cellulose digestible organic matter; ME, metabolisable energy; MJ, joules per mole.

Different letters within a column indicate significant differences ($P < 0.05$) between treatments. LSD multiple comparisons were used.

**Table 3** Effects of compost extract (CE), bio-organic manure (BOM) and CE + BOM (CEBOM) application on N, P, K contents of lucerne stems, leaves and soil.

| Treatment | Stem (g/kg) | | | Leaf (g/kg) | | | Soil (g/kg) | | |
|---|---|---|---|---|---|---|---|---|---|
| | N | P | K | N | P | K | N | P | K |
| Control | 2.30c | 0.72b | 11.90c | 3.77c | 1.30c | 7.51b | 0.81c | 0.18c | 1.31c |
| CE | 2.73b | 1.17a | 24.66b | 4.89b | 2.13b | 13.39a | 1.29b | 0.26b | 2.46b |
| BOM | 2.98b | 1.17a | 24.28b | 4.19b | 2.02b | 13.36a | 1.20b | 0.24b | 2.00b |
| CEBOM | 3.23a | 1.22a | 30.35a | 5.09a | 2.37a | 13.17a | 1.55a | 0.35a | 3.54a |

Notes.

Different letters within a column indicate significant differences ($P < 0.05$) between treatments. LSD multiple comparisons were used.

CEBOM >CE >BOM >control. The K content of stems and soil was significantly higher in the CEBOM treatment than in the other treatments, and both CE and BOM treatments had a higher K content in stems than the control treatment. Soil amendment treatments enhanced the K content of leaves in comparison with the control treatment. The K content of soil decreased in the order CEBOM >CE >BOM >control treatment.

**Table 4** Effects of compost extract (CE), bio-organic manure (BOM) and CE + BOM (CEBOM) application on MBC, MBN, MR, QCO$_2$ and numbers of bacteria and fungi in the rhizosphere soil of lucerne.

| Treatment | MBC (mg/kg) | MBN (mg/kg) | MR (mg/kg/d) | QCO$_2$ (mg C/kg) | Bacteria ($\times 10^6$ copies/ml) | Fungi ($\times 10^6$ copies/ml) |
|---|---|---|---|---|---|---|
| Control | 303.21c | 63.86c | 49.01c | 13.49c | 22.5d | 3.20c |
| CE | 384.44b | 70.60b | 57.33b | 14.92b | 34.7b | 4.33a |
| BOM | 340.73b | 62.80c | 59.81b | 14.36bc | 30.0c | 4.01b |
| CEBOM | 414.50a | 83.03a | 89.00a | 17.25a | 38.5a | 4.35a |

Notes.

MBC, microbial biomass Carbon; MBN, microbial biomass N; MR, microbial respiration rate; QCO$_2$, metabolic quotient, N= 20.
Different letters within a column indicate significant differences ($P < 0.05$) between treatments. LSD multiple comparisons were used.

**Table 5** Effects of compost extract (CE), bio-organic manure (BOM) and CE + BOM (CEBOM) application on soil characteristics.

| Treatment | pH | Bulk density (g/cm$^3$) | WHC (%) | OM (g/kg) | Alkaline N (mg/kg) | Available P (mg/kg) | Available K (mg/kg) |
|---|---|---|---|---|---|---|---|
| Control | 5.90b | 1.63a | 25.38bc | 14.22b | 67.60c | 4.02c | 79.88c |
| CE | 6.05ab | 1.47b | 28.48b | 23.03a | 102.46b | 35.85b | 159.79b |
| BOM | 5.95b | 1.46b | 25.87c | 22.53a | 110.63b | 50.08a | 204.70a |
| CEBOM | 6.33a | 1.32b | 35.17a | 25.37a | 128.20a | 52.83a | 214.37a |

Notes.

OM, organic matter; WHC, water holding capacity.
Different letters within a column indicate significant differences ($P < 0.05$) between treatments. LSD multiple comparisons were used.

## Biotic and abiotic characteristics of soils

MBC and MBN showed a significant increase with treatments (Table 4, Table S1). In the CEBOM treatment, MBC and MBN were 37% and 30% higher than in the control, respectively. MBC was not significantly different between CE and BOM treatments. Microbial respiration rates and metabolic quotients significantly increased in the order CEBOM >BOM = CE >control. Numbers of soil bacteria and fungi showed an increase in treatments with CE, BOM and CEBOM application, but were higher with CE alone than with BOM. The application of CE and BOM together strongly increased the amounts of bacteria in the rhizosphere soil of lucerne.

Of the three soil amendments, only CEBOM significantly increased soil pH and water holding capacity (Table 5, Table S1). The three amendment treatments significantly reduced bulk density and increased organic matter, soil alkaline-hydrolysable N, available P content and available K.

## Comparison between rhizobium nodulation among soil amendment treatments

The highest number of nodules and total nodule weight occurred in inoculated lucerne with the CE treatment (Fig. 2, Tables S1 and S2). The number of nodules and the total nodule weight of the inoculated lucerne showed no significant difference with non-inoculated

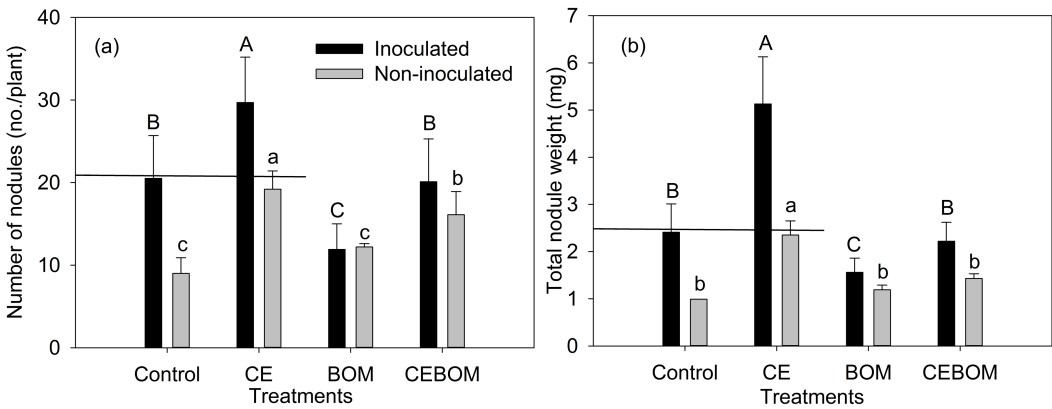

**Figure 2** **Effects of soil amendment treatments: control, compost extract (CE), bio-organic manure (BOM) and compost extract + bio-organic manure (CEBOM) on inoculated and non-inoculated nodulation of lucerne plants.** (A): number of nodules; (B): total nodule weight. Straight line showed the same level between inoculated control and non-inoculated CE adding treatment. Bars represent standard errors, $n = 20$.

lucerne under CE application. Under BOM application, the nodulation of lucerne decreased significantly. Table S3 showed the effects of treatments on plant N and soil N of inoculated lucerne.

## DISCUSSION

The shoot, root and total biomass and the nutritional value of inoculated and non-inoculated lucerne were significantly increased with the application of CE alone and CE with BOM together. It indicates that CE acts better than BOM for improving legume growth, because nitrogen- fixation can be suppressed by the addition of external nitrogen (*Goergen & Chambers, 2009*). However, it does not imply that BOM can substitute for CE, because highly synergistic effects on plant nutrient and soil properties were observed. Both soil biotic (MBC, MBN, MR) and abiotic (soil water holding capacity, organic matter, alkaline N, available P and K) factors were improved, and thus plant growth was enhanced.

When CE was applied, 164% higher plant biomass occurred, which could be mainly attributed to the stimulation of beneficial microorganisms (increased numbers of bacteria and fungi, Table 4) by accelerating the decomposition of organic materials and increasing the release of nutrients (*Javaid & Bajwa, 2011*). In contrast, BOM provided additional microorganisms and external nutrients to the soil (*Higa, 2001*). These inoculated microorganisms have to compete with naturally occurring bacteria (*Sherr, Sherr & McDaniel, 1992*). The interactions of different microorganisms may complement each other and further influence plant growth due to microbial diversity (*Van der Heijden, Bardgett & van Straalen, 2008*). By stimulating the proliferation of beneficial bacteria and restraining harmful microorganisms through CE addition, some essential substances such as nucleic acids, amino acids and bioactive substances (e.g., hormones and enzymes) are synthesized (*Javaid & Bajwa, 2011*). They can accelerate the decomposition of lignin materials in the soil and the mineralization of organic material as well as the control of soil diseases, resulting in

a change in soil microbial parameters and physical and chemical soil characteristics as seen in this study (*Kim et al., 2004*; *Javaid & Bajwa, 2011*). Besides, lucerne has been reported to produce root-derived antimicrobials in dole, terpenoid, benzoxazinone, flavonoid and isoflavonoid, which are used for inhibiting soil-borne pathogenic bacteria (*Dixon, 2001*) and thus enhance the symbiosis between plant and beneficial microorganisms. In soil organic amendment systems, beneficial microbes activated by CE and added by BOM accelerate the mineralization of soil organic matter (*Javaid, 2011*), which releases more nutrients for plant uptake (*Flores, Vivanco & Loyola-Vargas, 1999*).

Higher uptake of N, P and K by plant and higher soil content of N, P and K (Table 2) indicate that amendment treatments increased the availability of nutrients in the soil. In this study, the soil had a low pH value indicating acidic conditions, which resulted in low nutrient availability and low inorganic fertilization efficiency (*Chen et al., 2015*). The observed increase in soil pH with applying CE and BOM together demonstrated that a combination of soil conditioner and BOM could amend an acidic soil by accelerating the rate of microbial processes and thus releasing nutrients more quickly, as shown by other studies (*Hati et al., 2006*; *Lee et al., 2009*). Separately, they had no significant effects. CE includes less nutrient elements but has greater effects on soil and plant nutrient in comparison with BOM.

The identical numbers of nodules as well as nodule weight in inoculated and non-inoculated lucerne added with CE indicates the ability of CE for activating indigenous rhizobia (*Le et al., 2015*). Mechanisms for the maintenance of root-soil contact have been attributed to root exudation (*Le et al., 2015*). The plant-soil feedback driven by root exudates could initiate and manipulate a signal exchange between roots and soil organisms (*Keymer & Lankau, 2017*). CE may stimulate root exudates, which could initiate one-way signals informing roots on chemical and physical soil properties, and regulating the symbiotic and protective interactions with microbes (*Jones, Farrar & Giller, 2003*; *Johnson et al., 2015*). The exudates accumulate mainly inducible antimicrobial compounds gathered in the roots (*Flores, Vivanco & Loyola-Vargas, 1999*; *Dixon, 2001*). Many indigenous microorganisms in the soil are not able to interact synergistically with plant species; therefore, a number of antimicrobial root exudates are secreted. *Le et al. (2015)* confirmed that some endophytic actinobacteria had the potential to enhance the growth of lucerne and its interactions in rhizobial symbiosis. The mechanism by which CE produces a stimulatory effect on properties soil and plant growth needs to be further explored.

## CONCLUSIONS

This study demonstrated that CE application alone and in combination with BOM enhanced lucerne biomass and nutritional value by improving nodulation and soil biotic and abiotic properties. The similar effects of CE and BOM on plants and soils does not necessary mean that they can be treated as substitutes for one another. Rather, their combination provides a mode of soil amendment which improves soil quality more efficiently. CE synthesizes some kinds of active substrates that stimulate the activity of

indigenous rhizobia in the soil rather than adding extra microorganisms into the soil as BOM does. Root exudation may help in regulating soil properties and root-microbe-soil interactions. However, the nature of their reactions in the soil need to be further explored.

## ACKNOWLEDGEMENTS

We acknowledge all students for helping sampling measurement. We acknowledge Nanjing Agriculture University for providing lab facilities.

### Funding

This work was supported by the Postdoctoral Fund in Jiangsu Province (1601036A) and the Chinese Postdoctoral Science Foundation (1061265C). The funders had no role in study design, data collection and analysis, decision to publish, or preparation of the manuscript.

### Grant Disclosures

The following grant information was disclosed by the authors:
Postdoctoral Fund in Jiangsu Province: 1601036A.
Chinese Postdoctoral Science Foundation: 1061265C.

### Competing Interests

The authors declare there are no competing interests.

### Author Contributions

- Haiyan Ren conceived and designed the experiments, performed the experiments, analyzed the data, contributed reagents/materials/analysis tools, wrote the paper, prepared figures and/or tables, reviewed drafts of the paper.
- Jian Hu conceived and designed the experiments, contributed reagents/materials/analysis tools, reviewed drafts of the paper.
- Yifei Hu performed the experiments, analyzed the data, prepared figures and/or tables.
- Gaowen Yang analyzed the data, contributed reagents/materials/analysis tools, reviewed drafts of the paper.

### Data Availability

 The raw data is provided as a Supplementary File.

### Supplemental Information

Supplemental information for this article can be found online at http://dx.doi.org/10.7717/peerj.3775#supplemental-information.

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
