# Peer review of "Divergence of compost extract and bio-organic manure effects on lucerne plant and soil"

_PeerJ, doi:10.7717/peerj.3775_

## Round 0.1 · original submission · Major Revisions

· Academic Editor

Major Revisions

The paper has been reviewed favourably by 2 reviewers. However I have several concerns on the experimental design, in particular the time and period of OM and CE application:

-the seedlings were grown only for 8 days (Line 110). The results were then proved to be dubious as I would not expect any difference in 8 days of seedlings growth
- It is unclear how and why the soil mixture is determined? Why do you mix soil with sand? What is the particle-size distribution of the mixture soil?
- It is unclear how long or incubation period for the application of OM and CE in the soil?
- I am not clear why application of CE can increase soil C more than addition of OM? The amount of OM in CE is much less than OM addition? What is the C content of CE?
Table 5: The application rate of OM is 5 g/kg, and why is the increase in OM (compared to control) larger than 5 g/kg (14 to 22 g/kg)?
- What is the mechanism that caused bulk density to decrease with application of CE? It does not make sense as bulk density is mainly controlled by particulate organic matter.

In addition, the paper's English is unclear, it needs to be fully revised so it can be fully comprehended by international readers.

Reviewer 1 ·

Basic reporting

Topic of the paper is still a subject of investigations and it is suitable for the PeerJ journal.
This manuscript is properly organized and written clearly but I suggest to improve grammar and syntax.
Literature references for methodology for determination of: soil organic matter content, available P and K, content of total C used in the paper are old and currently not used.

Experimental design

Title should be rewriting because in my opinion 'organic manure' does not exist, the origin of manure should be given eg. cow manure. Consequently, it should be changed throughout the manuscript (abstract, keywords).
Methodology is too vague and without important details. The experimental methods are not sufficiently described. Important information is not provided, such as e.g. the characteristics of organic manure used in the experiment.

Validity of the findings

The paper includes valuable results, but it requires minor changes in terms of: terminology, before publishing.

Additional comments

The manuscript is interesting, it concerns an important problem and has considerable scientific value.
Detailed comments for tables are provided directly in the text.
Other specific objections that I have seen during reading the article are presented below.
Line 3-6: this sentence should be rewritten;
Line 36: "level' I suggest changing to 'content';
Line 69: 'lab' for laboratory;
Lines 197, 240, 242: 'PH' for pH.

·

Basic reporting

The experiment was well-conducted, with a clearly explained and justified experimental design. Data were analysed with adequate statistical methods. Results were straightforward and clearly presented, with tables and figures (including supplementary material) all necessary. The discussion was based on updated literature and is of utmost interest for the future of agriculture in China and everywhere compost extracts and organic manure are available for the development of sustainable agriculture, and need to be compared for the improvement of plant growth and soil condition.

Experimental design

Nothing to add

Validity of the findings

Nothing to add

Additional comments

Minor changes have been suggested in the annotated manuscript. I paste below some additional comments done in the margins:
• Line 84, “It was made of pig dung compost and amino acid fertilizer”: What was the proportion of both components?
• Line 197, “CE + OM”: It would be desirable to have a unique designation for the combined treatment, either CE + OM or CEOM. Please chose one of them and apply it to the whole text, tables and figures.
• Lines 227-228, “some essential substances such as nucleic acids, amino acids and bioactive substances (e.g. hormones and enzymes) are synthesized”: A reference is needed.
• Lines 258-259, “They accumulate mainly inducible antimicrobial compounds gathered in the roots”: Please reword this sentence, which is quite unclear.

---

## Round 0.2 · accepted · Accept

· Academic Editor

Accept

Thank you for addressing the recommended changes. I agree with the reviewer recommendation that the work is now ready publication

·

Basic reporting

All changes I suggested have been taken into account. For my part, I think that the paper is now ready for publication in PeerJ. No further changes are requested.

Experimental design

OK

Validity of the findings

OK

Additional comments

OK